# Fetal and neonatal echocardiographic analysis of biomechanical alterations for the systemic right ventricle heart

Brett A. Meyers[1], Sayantan Bhattacharya[2], Melissa C. Brindise[3], Yue-Hin Loke[4], R. Mark Payne[5], Pavlos P. Vlachos[1] *

1 School of Mechanical Engineering, Purdue University, West Lafayette, IN, United States of America, 2 Department of Mechanical Engineering, University of Maryland, Baltimore County, Baltimore, MD, United States of America, 3 Department of Mechanical Engineering, Pennsylvania State University, University Park, PA, United States of America, 4 Department of Cardiology, Children's National Hospital, Washington, District of Columbia, United States of America, 5 Division of Pediatric Cardiology, Department of Pediatrics, Indiana University School of Medicine, Indianapolis, IN, United States of America

* pvlachos@purdue.edu

## Abstract

### Background

The perinatal transition's impact on systemic right ventricle (SRV) cardiac hemodynamics is not fully understood. Standard clinical image analysis tools fall short of capturing comprehensive diastolic and systolic measures of these hemodynamics.

### Objectives

Compare standard and novel hemodynamic echocardiogram (echo) parameters to quantify perinatal changes in SRV and healthy controls.

### Methods

We performed a retrospective study of 10 SRV patients with echocardiograms at 33-weeks gestation and at day of birth and 12 age-matched controls. We used in-house developed analysis algorithms to quantify ventricular biomechanics from four-chamber B-mode and color Doppler scans. Cardiac morphology, hemodynamics, tissue motion, deformation, and flow parameters were measured.

### Results

Tissue motion, deformation, and index measurements did not reliably capture biomechanical changes. Stroke volume and cardiac output were nearly twice as large for the SRV compared to the control RV and left ventricle (LV) due to RV enlargement. The enlarged RV exhibited disordered flow with higher energy loss (EL) compared to prenatal control LV and postnatal control RV and LV. Furthermore, the enlarged RV demonstrated elevated vortex strength (VS) and kinetic energy (KE) compared to both the control RV and LV, prenatally and postnatally. The SRV showed reduced relaxation with increased early filling velocity ($E$)

**Data Availability Statement:** All relevant data are available within the paper and its Supporting Information files.

**Funding:** PPV and RMP received support for this project by the Indiana Clinical and Translational Sciences Institute (https://indianactsi.org) and funded, in part, by the National Center for Advancing Translational Sciences, Grant UL1TR002529, Clinical and Translational Sciences Award (https://ncats.nih.gov/ctsa) and, in part, by the Eunice Kennedy Shriver National Institute of Child Health and Human Development, Grant 1R21HD109490 (https://www.nichd.nih.gov). The content is solely the responsibility of the authors and does not necessarily represent the official views of the National Institutes of Health. No industry partnerships collaborated on or funded this work. The funders had no role in study design, data collection and analysis, decision to publish, or preparation of the manuscript.

**Competing interests:** I have read the journal's policy and the authors of this manuscript have the following commpeting interests: BAM, MCB, and PPV have intellectual property filings for each analysis algorithm in the analysis method used in this manuscript. BAM and PPV are involved with Cordian Technologies, a start-up founded by PPV, which licenses the technologies used in this work. RMP has a significant interest in Larimar Therapeutics, Inc., which is unrelated to this work. He also has funding from NHLBI as a co-investigator (1P01HL134599), and as site PI on an NHLBI consortium project (UG1HL135678). YHL receives partial salary support from NHLBI (R21-HL156045-01)

**Abbreviations:** AV, = Atrioventricular; ALAX, = Apical long axis; CFI, = color flow imaging; CO, = Cardiac output; DoVeR, = Doppler velocity reconstruction; EL, Energy loss; GLS, Global longitudinal strain; IVPD, Intraventricular pressure difference; LV, Left ventricle; RV, Right ventricle; SRV, Systemic right ventricle; SV, Stroke volume; VS, Vortex strength; $\Delta P$, Pressure difference * A full list of abbreviations is provided in the S1 Table.

compared prenatally to the LV and postnatally to the control RV and LV. Furthermore, increased recovery pressure ($\Delta P$) was observed between the SRV and control RV and LV, prenatally and postnatally.

## Conclusions

The novel hydrodynamic parameters more reliably capture the SRV alterations than traditional parameters.

## Introduction

The perinatal transition induces significant hemodynamic changes in newborns, involving four major events: loss of umbilical cord blood flow, first breath, circulatory shift, and myocardial performance changes [1]. Subsequently, the lungs become the primary respiratory organ, necessitating fluid clearance, leading to decreased systemic venous return and increased systemic resistance [2]. Left ventricle (LV) and right ventricle (RV) hemodynamics show a marked increase in cardiac output (CO)–LV output rises by 300%, and RV output by 150% [3], resulting from alterations in preload, afterload, and pericardial pressure effects.

In both systemic right ventricles (SRVs), which is a critical congenital heart defect (CHD), and normal two-ventricle hearts, changes in pericardial pressures affect ventricular expansion during diastolic filling, influencing blood flow patterns, also known as flow fields. Current guidelines primarily focus on capturing systolic function measurements, e.g., stroke volume, fractional area change, tricuspid annular plane systolic excursion, or tissue Doppler-derived peak systolic annular velocity (s'). However, these measurements are often inaccurate, highly variable, and influenced by factors such as image quality, probe placement, heart size, and tissue motion [4, 5]. Consequently, there is a lack of clinical understanding regarding the assessment of diastolic function in the RV, which undergoes significant alterations in the perinatal period and is crucial for SRV physiology [6, 7].

Little is known regarding these flow fields and their effects on cardiac hemodynamics in both prenatal and postnatal states. Initial work employed 4D flow MRI to examine cardiac chamber flow in newborns [8]. While bulk flow features were reliably observed, finer features typical in these flows remained unresolved. Subsequent studies on congenital heart defects (CHDs) using 4D flow MRI are scarce [9] but no additional studies of this kind exist. However, the evolving potential of the fetal 4D flow MRI [10, 11], offering higher resolution voxel sizes, may address this gap. Still, it is difficult to perform fetal 4D flow MRI due to uncontrollable factors, including fetal motion, breathing, and positioning of the mother and fetus. Recently, echocardiography-based methods explored cardiac flow in fetal and neonate patients, utilizing blood speckle imaging [12] or color Doppler flow reconstruction [13]. Although demonstrated for feasibility, none have explored their potential for understanding diastolic function across the perinatal period. Quantitative tools can resolve flow-induced vortices, energy losses, and pressure distributions, characterizing abnormal flow patterns in SRV hearts [14]. We have recently carried out a similar study in tetralogy of Fallot patients [15].

In this current study, we applied an integrated and automated echocardiography analysis method for measuring cardiac biomechanics from fetal and neonatal echocardiograms to quantify standard and novel hemodynamics biomarkers. Our goal was to apply novel, integrated echo-based imaging tools to better understand the hemodynamic changes in the SRV during the transition from fetal to postnatal life. The analysis collected chamber, annular

motion, strain, and hydrodynamics parameters. The tools employed are vendor-agnostic and do not rely on heuristics adopted from adult echocardiography. These advancements enable conventional and novel biomechanics measurements to be robustly collected from fetal and neonatal echocardiograms.

Our study explored how biomechanics parameters quantified from echocardiography change from late gestation (> 28 weeks gestational age) and day of birth for single ventricle subtype SRVs and the healthy left (LV) and RV. Here, we compared standard and novel echo-based approaches to better understand the perinatal changes in the SRV and normal heart. We hypothesize that the single ventricle subtype SRV has altered diastolic flow compared to both ventricles of the healthy heart, captured by the biomechanics parameters. The SRV presents a challenging clinical scenario that emphasizes the need for robust parameters to understand patient cardiac health comprehensively.

## Methods

### Study cohort

Patient examinations were retrospectively selected from within the Indiana University Health and Children's National Hospital networks between July 1st, 2017, and April 30th, 2022. The cohort comprised ten single ventricle subtype SRV subjects with fetal echocardiography performed at 33 weeks average gestational age and with pediatric transthoracic echocardiography at day of birth. Twelve age-matched healthy controls were included. Datasets without B-mode and CFI recordings in the apical long-axis (ALAX) view were excluded. All exams were deidentified before the data sharing between institutions for analysis. The Institutional Review Board for Human Studies for Purdue University (IRB-2021-64), Children's National Hospital (Pro00010769), and IU (1904623285) approved this study, which was exempt from informed consent.

### Echocardiography

Sonographers performed fetal and pediatric echocardiograms on one of either Acuson SC2000 ultrasound systems (Siemens Medical Solutions USA, Inc., Malvern, Pennsylvania), iE33/Epic 7 (Philips, Andover, Massachusetts), or Vivid E-95 (General Electric, Boston, MA, USA) ultrasounds. American Society of Echocardiography guidelines were followed [16, 17]. ALAX B-mode and CFI acquisitions were obtained with appropriate Nyquist limit and color box covering the entire ventricular cavity. Frame rates varied between the fetal and pediatric sessions due largely to differences in imaging depths. The median, maximum, and minimum values are reported in Table 1.

### Image analysis workflow

The analysis workflow, summarized in Fig 1, outputs ventricular cardiac biomechanics measurements from B-mode and CFI ALAX recordings. These modalities were utilized because

**Table 1. Frame rates for apical four-chamber scans within the cohort.**

| | | B-mode Echocardiography | | | Color Doppler Echocardiography | | |
|---|---|---|---|---|---|---|---|
| Frame rate (FPS) | | Minimum | Median | Maximum | Minimum | Median | Maximum |
| SRV | Prenatal | 22 | 77 | 85 | 17 | 21 | 62 |
| | Postnatal | 40 | 76 | 101 | 19 | 29 | 40 |
| CTRL | Prenatal | 42 | 66 | 87 | 15 | 18 | 24 |
| | Postnatal | 30 | 30 | 95 | 18 | 27 | 30 |

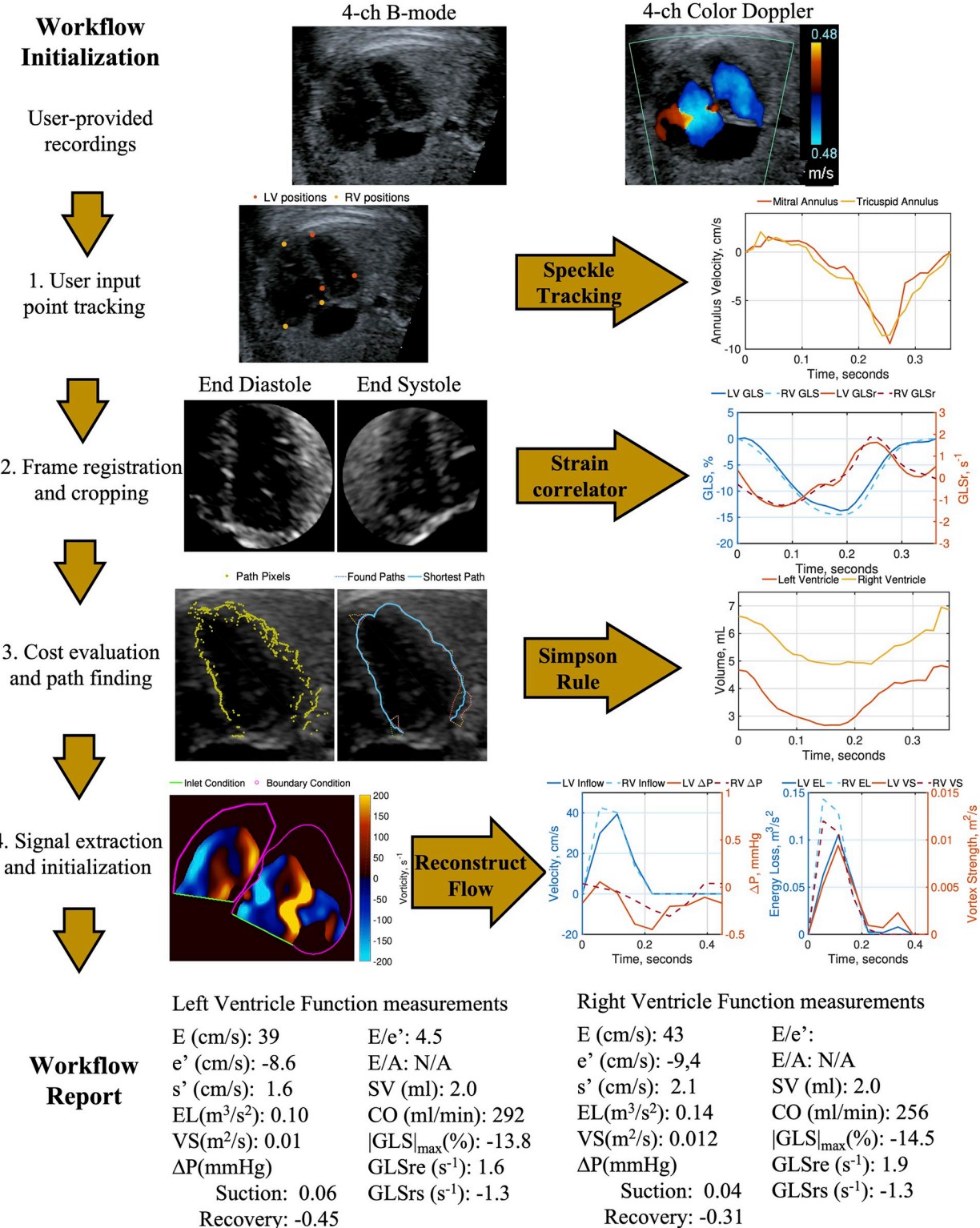

**Fig 1. Echocardiogram analysis workflow.** Analysis begins with the user providing 4C views. (1) AV annulus and apex feature points are provided to initialize automated analysis. (2) B-mode frames are co-registered, cropped, and processed to quantify GLS. (3) B-mode frames are evaluated to find pixel costs and paths for ventricle segmentation and volume quantification. (4) Color Doppler frames are processed to extract the signal, segment the ventricle, set initial conditions, and reconstruct velocity fields. Cardiac function measurements are compiled into a workflow report.

the sonographers are well-trained in their recording, which enhances analysis consistency. The workflow automates measurements, enabling challenging and highly user-variable analysis to become routine. All algorithms were run in MATLAB (The MathWorks, Natick, Massachusetts).

**Step 1: Tracking user input annulus and apex positions.** A single set of user inputs marking the ventricle apex and atrioventricular (AV) annulus positions on the initial recorded frame, depicted as the red and yellow positions in Fig 1-1 is required per scan. The input AV annulus positions are located at the annulus-septal wall and annulus-lateral wall junctions. These positions are tracked temporally using a speckle tracking algorithm based on the Fourier domain cross-correlation [18, 19]. A more detailed description of the tracking algorithm is provided in the S1 Text.

The tracked positions provide measurements of ventricle relaxation that occurs during diastole and contraction that occurs during systole [20], driven by the AV annulus. Peak annulus velocities for systolic ejection (s') and early diastolic filling (e') are automatically measured from the velocity time-series, also depicted in Fig 1-1, obtained through the tracking process using peak finding. Automated speckle tracking mitral annulus position and velocity measurements has been previously validated with adult patients [21].

**Step 2: Global longitudinal strain.** A novel algorithm is used to measure global longitudinal strain (GLS) from the whole ventricle image [22]. This algorithm overcomes limitations for strain estimation associated with manual tracings of the chambers [4, 14] and the use of speckle pattern matching kernels [23, 24], which are sensitive to image spatial and temporal resolution. B-mode recording frames are co-registered using tracked positions and the ventricle image is isolated, shown in Fig 1-2. For co-registered frame pairs containing the ventricle, a specialized logarithm-scaled kernel estimates global longitudinal strain rate (GLSr) between frames. The GLSr estimates are integrated to resolve GLS. Peak GLS (|GLS|, max) is output to quantify deformation from the GLS timeseries using peak finding. A more detailed description of the GLS algorithm is provided in the S1 Text.

**Step 3: Unsupervised chamber segmentation.** The unsupervised segmentation algorithm (ProID) automates ventricle detection and volume estimation [25]. The algorithm identifies ventricle boundaries using a machine vision algorithm [26] that finds the shortest path of pixels around the ventricle image, shown in Fig 1-3. ProID addresses contrast-to-noise and resolution issues common in natal imaging [14] with an echocardiogram-specific cost-matrix. The tracked positions from **Step 1** are used to initialize ProID for each frame. A more detailed description of the ProID algorithm is provided in the S1 Text.

Segmentation-derived volumes are computed from the identified boundaries using Simpson's rule. Previous research demonstrated the accuracy of Simpson's rule in estimating actual volume using animal models in fetal echocardiography [27]. End-diastolic volume (EDV) was measured at the initial volume of each cycle in the processed volume time-series. End-systolic volume (ESV) was determined using peak finding for the minimum volume in the cardiac cycle. Stroke volume (SV), calculated as the difference between EDV and ESV for each cycle, and cardiac output (CO), representing the stroke volume multiplied by heart rate, served as output measures for systolic function.

**Step 4: Color flow imaging hemodynamics analysis of diastolic flow.** Doppler vector reconstruction (DoVeR) resolves the underlying 2D velocity vector field of blood flow in the ventricles from color Doppler imaging using the relationship between flow rate and fluid rotation [28]. DoVeR uses the tracked positions and ProID to segment the ventricle in each frame. These segmentations are used to set boundary conditions for the DoVeR algorithm, as shown in Fig 1-3. A more detailed description of the DoVeR algorithm is provided in S1 Text.

The DoVeR velocity vector fields of ventricular blood flow, denoted as $\overrightarrow{u}$, are evaluated to quantify biomechanics parameters, including peak early filling velocities (E), kinetic energy (KE), energy loss (EL), and vortex strength (VS) as well as the annulus-to-apex recovery pressure difference (recovery $\Delta P$) from the intraventricular pressure difference (IVPD) and AV valve center to minimum pressure distance (AV-to-$P_{min}$). The E is measured by peak finding from the velocity time-series for each cardiac cycle. The KE, a measure of the amount of energy blood flow has due to its motion, is a computed integral of the summed square, $\overrightarrow{u}^2$, from the velocity measurements over the ventricle area,

$$KE = \int_A \frac{1}{2}\rho \sum \overrightarrow{u}^2 dS. \tag{1}$$

Here, $\rho$ is the density of blood, assumed as 1030 $kg/m^3$. The EL, a measure of the KE lost due to rotation, acceleration, or deceleration, is computed by integrating the spatial gradients for each velocity measurement over the ventricle area,

$$EL = \int_A \frac{1}{2}\mu \sum_{ij} \left( \frac{\partial u_i}{\partial x_j} + \frac{\partial u_j}{\partial x_i} \right)^2 dS. \tag{2}$$

Here, $\mu$ is the blood viscosity which is assumed as 3 $mPa{\cdot}s$, and $i,j$ are indices representing the spatial components of the velocity vector field in the horizontal and vertical directions. Vortex strength (VS), an absolute quantity of the total amount of rotation observed in the flow field, is quantified from vorticity ($\overrightarrow{\omega}$), the potential for blood flow rotation, integrated over the ventricle area. Vorticity is computed taking the curl of the velocity field $\overrightarrow{u}$,

$$\overrightarrow{\omega} = \nabla \times \overrightarrow{u}. \tag{3}$$

Thus, VS is computed as,

$$VS = \int_A |\overrightarrow{\omega}| \cdot dS. \tag{4}$$

Relative pressure of the DoVeR blood flow velocity vector fields, is estimated by integration of the pressure gradients [29] which are computed based on the Navier Stokes Equation (NSE),

$$\frac{\partial P}{\partial x_i} = -\rho \left( \frac{\partial u_i}{\partial t} + u_j \frac{\partial u_i}{\partial x_j} \right) + \mu \frac{\partial^2 u_i}{\partial x_j \partial x_j}. \tag{5}$$

The IVPD is calculated as the difference between pressure at the AV ($P_{TV}$) and the ventricle ($P_{Apex}$) such that,

$$IVPD = \Delta P = P_{TV} - P_{Apex}. \tag{6}$$

## Statistical analysis

Data are reported as the median with inter-quartile range (IQR). We compared each parameter across conditions using the Kruskal-Wallis test; this test is non-parametric, does not assume normal distribution, and looks to determine if the distributions are significantly different based on the median. Post-hoc evaluation using Tukey honest significant difference was performed to obtain the p-value for each pair of tested conditions. A two-tailed p-value $< 0.05$ was considered statistically significant. We performed statistical analysis using the MATLAB Statistics toolbox. Additional metrics computed but not presented are provided in S2 Table.

**Table 2. Demographics of systemic right ventricle subjects used in this study.**

| ID | Sex | Weight (kg) | Anatomical Subtype | Complications |
|----|-----|-------------|--------------------|---------------|
| 01 | M | 3.0 | Mitral Atresia | Restrictive PFO |
| 02 | M | 3.7 | Mitral Atresia | None |
| 03 | M | 3.2 | Mitral Atresia | Dextrocardia, L-TGA, Pulmonary Atresia, interrupted IVC with bilateral SVC |
| 04 | F | 3.7 | Mitral Atresia | L-TGA, Pulmonary Atresia, bilateral PDA, Heterotaxia with left IVC and SVC |
| 05 | F | 3.4 | Mitral Atresia | None |
| 06 | M | 3.6 | Mitral Stenosis | Cardiogenic Shock |
| 07 | F | 2.5 | Mitral Stenosis | None |
| 08 | F | 3.0 | Mitral Atresia | Total anomalous pulmonary venous drainage |
| 09 | M | 2.6 | Mitral Atresia | Moderate tricuspid valve insufficiency |
| 10 | M | 2.6 | Mitral Stenosis | Mild tricuspid valve insufficiency |

M indicates male; F, Female; PFO, Patent Foramen Ovale; L-TGA, L-looped transposition of the great arteries; IVC, Inferior vena cava; SVC, Superior vena cava; PDA, Patent ductus arteriosus

## Results

### Subject demographics

Relevant clinical information for each of the 10 SRV subjects are provided in Table 2. The cohort composed of 6 males and 4 females. Mitral atresia was the most common subtype, affecting 7 patients, followed by 3 patients with mitral stenosis.

### Systolic parameters in prenatal SRV and controls

Measured parameters for fetal control and SRV hearts are provided in Table 3. Major differences were observed for morphology and systolic parameters. SV (ml) and CO (ml/min) were elevated for the SRV compared to the control LV by a nearly two-fold statistically significant difference but not the RV. Peak s' (cm/s) was comparable for the SRV and both control ventricles. The SRV Peak GLS was elevated compared to the control LV but not the RV.

### Diastolic parameters in prenatal SRV and controls

Major differences were observed for several of the diastolic parameters. SRV E (cm/s) was significantly elevated compared to the control LV but not the RV. The E/e' quantity was elevated–without significance–in the SRV compared to the control RV and LV. Conversely, the E/A quantity was comparable for the SRV against both the control ventricles. Mean flow EL (FEL), a measurement of total EL over the ventricle volume in the SRV (mW/m) averaged over diastole, was significantly different compared to both the control RV and control LV. Similarly, mean KE (mJ/m) and mean VS in the SRV ($cm^2$/s) were significantly different compared to both the control RV and LV. Recovery $\Delta P$ was significantly different in the SRV compared to the control RV and LV. Suction $\Delta P$ was significantly different in the SRV compared to the control RV but not the control LV. AV-to-$P_{min}$ occurred significantly further from the annular plane for the fetal SRV compared to the control RV and LV.

### Qualitative assessment between prenatal SRV and controls

Representative changes in ventricular volumes, strains, and intracardiac flows are illustrated in Fig 1 for a fetal SRV vs. fetal control. The SRV exhibits several major differences compared to the control RV. First, the SRV volume is larger (Fig 2A) and has an altered AV valve position, producing an asymmetric vortex pair during diastole (Fig 2C and, 2D), with the free wall

**Table 3. Echocardiographic measurements obtained from automated analysis platform.**

| | | SRV (n = 10) | CTRL LV (n = 12) | | CTRL RV (n = 12) | |
|---|---|---|---|---|---|---|
| | | Median (IQR) | Median (IQR) | p-value | Median (IQR) | p-value |
| Heart rate | Prenatal | 141(136, 162) | 140(137, 145) | 0.283 | - | - |
| (bpm) | Postnatal | 153(138, 160) | 119(106, 163) | 0.322 | - | - |
| | p-value | 1.000 | 0.429 | | - | - |
| **Systolic Parameters** | | | | | | |
| Stroke Volume | Prenatal | 2.91(2.46, 4.30) | 1.56(1.33, 2.02) | **0.001** | 2.47(1.83, 3.41) | 0.129 |
| (ml) | Postnatal | 3.70(2.86, 4.59) | 2.83(2.52, 3.67) | 0.187 | 2.59(1.24, 3.44) | 0.129 |
| | p-value | 0.545 | **0.001** | | 0.863 | |
| Cardiac Output | Prenatal | 435(377, 539) | 233(209, 299) | **0.002** | 341(288, 439) | 0.166 |
| (ml/min) | Postnatal | 539(398, 610) | 424(265, 583) | 0.235 | 327(159, 471) | 0.075 |
| | p-value | 0.650 | **0.024** | | 0.603 | |
| $\|GLS\|_{max}$ | Prenatal | 18.7(15.1, 25.1) | 15.8(13.5, 17.0) | 0.138 | 17.8(16.2, 18.8) | 0.843 |
| (%) | Postnatal | 16.9(15.4, 19.2) | 17.9(16.8, 20.7) | 0.235 | 23.4(19.5, 25.8) | **0.008** |
| | p-value | 0.427 | **0.020** | | **0.011** | |
| s' | Prenatal | 2.73(2.10, 3.59) | 2.70(2.01, 3.55) | 0.843 | 3.30(1.99, 4.29) | 0.391 |
| (cm/s) | Postnatal | 2.92(2.32, 4.09) | 2.45(1.74, 2.67) | 0.065 | 2.98(2.47, 3.38) | 0.742 |
| | p-value | 0.290 | 0.204 | | 0.686 | |
| **Diastolic Parameters** | | | | | | |
| e' | Prenatal | 2.85(2.44, 4.45) | 3.50(3.03, 4.15) | 0.644 | 4.33(3.45, 5.49) | 0.129 |
| (cm/s) | Postnatal | 4.03(349, 5.22) | 2.98(2.55, 3.85) | 0.138 | 3.52(2.75, 4.64) | 0.510 |
| | p-value | 0.290 | 0.564 | | 0.326 | |
| E | Prenatal | 44.8(34.4, 48.0) | 29.7(28.2, 34.5) | **0.027** | 37.3(35.0, 38.9) | 0.114 |
| (cm/s) | Postnatal | 62.1(60.6, 68.0) | 41.1(36.5, 45.8) | **0.003** | 34.9(30.5, 43.3) | **<0.001** |
| | p-value | **0.001** | **0.030** | | 0.436 | |
| E/e' | Prenatal | 14.8(7.1, 23.3) | 8.6(7.6, 11.2) | 0.210 | 9.9(5.7, 11.6 | 0.129 |
| | Postnatal | 15.4(13.3, 17.9) | 13.2(7.7, 16.8) | 0.262 | 9.9(7.9, 11.0) | **0.004** |
| | p-value | 0.762 | 0.312 | | 0.773 | |
| E/A | Prenatal | 1.26(1.14, 1.37) | 1.32(1.25, 1.47) | 0.468 | 1.40(1.16, 3.11) | 0.210 |
| | Postnatal | 1.41(1.16, 1.82) | 1.16(0.92, 1.44) | 0.086 | 1.16(0.89, 2.23) | 0.235 |
| | p-value | 0.272 | 0.141 | | 0.174 | |
| Kinetic Energy | Prenatal | 27.5(21.9, 29.8) | 4.2(3.6, 5.1) | **<0.001** | 8.2(7.0, 10.6) | **<0.001** |
| (mJ/m) | Postnatal | 25.7(19.3, 33.2) | 10.8(4.0, 13.8) | **0.002** | 5.0(4.4, 6.1) | **<0.001** |
| | p-value | 0.880 | **0.050** | | **0.017** | |
| Flow energy loss | Prenatal | 23.2(11.2, 36.3) | 4.9(3.1, 6.1) | **0.003** | 8.3(5.9, 10.4) | **0.048** |
| (mW/m) | Postnatal | 31.8(23.5, 39.9) | 10.8(6.5, 15.5) | **0.002** | 8.2(6.1, 9.1) | **<0.001** |
| | p-value | 0.199 | **0.013** | | 0.686 | |
| Vortex Strength | Prenatal | 248(197, 326) | 97(77, 102) | **<0.001** | 135(123, 145) | **0.001** |
| (cm²/s) | Postnatal | 315(258, 342) | 180(121, 263) | **0.005** | 132(98, 150) | **<0.001** |
| | p-value | 0.140 | **0.004** | | 0.686 | |
| Suction ΔP | Prenatal | 0.31(0.19, 0.36) | 0.24(0.21, 0.30) | 0.229 | 0.18(0.07, 0.19) | **0.034** |
| (mmHg) | Postnatal | 1.27(0.81, 1.46) | 0.54(0.33, 0.64) | **0.001** | 0.67(0.41, 0.72) | **0.006** |
| | p-value | **<0.001** | **<0.001** | | **<0.001** | |
| Recovery ΔP | Prenatal | $-1.43\binom{-1.96,}{-0.89}$ | $-0.42\binom{-0.50,}{-0.34}$ | **<0.001** | $-0.55\binom{-0.62,}{-0.47}$ | **0.002** |
| (mmHg) | Postnatal | $-2.26\binom{-2.46,}{-1.97}$ | $-0.73\binom{-1.06,}{-0.49}$ | **0.002** | $-0.38\binom{-0.68,}{-0.31}$ | **<0.001** |
| | p-value | **0.007** | **0.005** | | 0.184 | |

(*Continued*)

**Table 3.** (Continued)

| | | SRV (n = 10) | CTRL LV (n = 12) | | CTRL RV (n = 12) | |
|---|---|---|---|---|---|---|
| | | Median (IQR) | Median (IQR) | p-value | Median (IQR) | p-value |
| Min. ΔP Loc. | Prenatal | 9.2(7.5, 10.0) | 4.1(3.8, 4.4) | **0.001** | 4.8(3.9, 5.3) | **0.004** |
| (mm) | Postnatal | 4.9(4.3, 5.7) | 4.4(2.8, 5.7) | 0.428 | 2.9(2.3, 3.5) | **0.007** |
| | p-value | **0.003** | 0.862 | | **0.028** | |

e' indicates systolic annular velocity; |GLS|$_{max}$, peak absolute global longitudinal strain; e', early diastolic annular velocity; E, early diastolic filling velocity; E/A, early-to-late diastolic filling velocity ratio; ΔP, Pressure difference. P-values reported in labeled columns are computed against SRV values for the current age; those reported in labeled rows are computed between age groups.

vortex occupying a larger area than the septal wall vortex. Second, an augmented pressure field was observed during both early diastole (Fig 2C-1), and late diastole (Fig 2C-2), where the free wall vortex had stronger low pressure and the apex had stronger high pressure compared to the healthy heart. Third, stronger EL was observed during both early diastole (Fig 2D-1) and late diastole (Fig 2D-2) compared to the healthy heart due to more disordered flow, which produces greater shear. Fourth, the SRV time-series (Fig 2C and 2D) did not show distinctly separate early and late diastolic filling; instead, these phases were fused. Thus, the fetal SRV experienced stronger reversal IVPD and FEL due to altered filling patterns.

## Systolic parameters in postnatal SRV and controls

Measured parameters for the neonate SRV hearts and controls are provided in Table 3. Major differences were observed for morphology as well as for systolic parameters. Observed volume changes occurred for both the SRV and controls, but separation of SV and CO from the control LV and RV was not observed, although CO was trending toward significance for the RV ($p = 0.075$). SRV s' was elevated and trending toward significance compared to the control LV ($p = 0.065$) but not the RV. Peak GLS for the SRV was significantly depressed compared to the control RV but not the LV.

## Diastolic parameters in postnatal SRV and controls

Major differences were observed for several of the same diastolic parameters for neonates' hearts as fetal hearts. Comparable peak e' (cm/s) was observed between the SRV and the controls. SRV E remained significantly elevated compared to the control RV and LV. SRV E/e' remained elevated compared to the control RV but not the LV. SRV E/A increased compared to the control RV and LV, trending toward significance for the latter case ($p = 0.086$). Mean KE, mean FEL, mean VS, recovery ΔP, and suction ΔP each were significantly increased for the SRV compared to the control LV and RV. AV-to-P$_{min}$ decreased for both the SRV and control RV but remained unchanged for the LV, with statistical significance observed between the SRV and control RV.

## Qualitative assessment between postnatal SRV and controls

Representative changes in ventricular volumes, strains, and intracardiac flows are illustrated in Fig 3 for a neonate SRV and a control. The major differences observed for the SRV are more prevalent after birth. The increased volume and altered AV valve position for the SRV produced an asymmetric vortex pair during diastole (Fig 3C and 3D) with the free wall vortex occupying a larger area than the septal wall vortex. An augmented pressure field was again

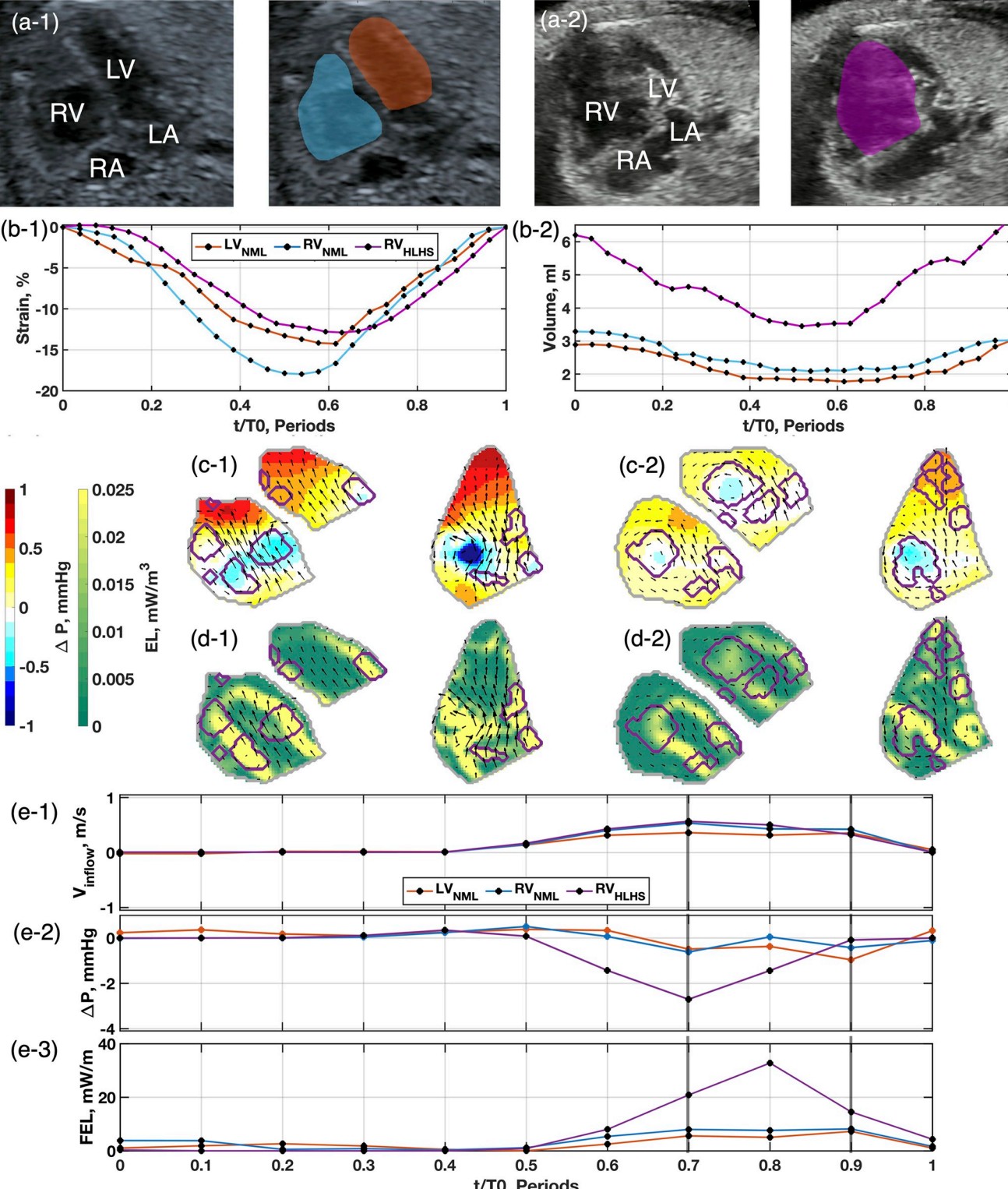

**Fig 2. Comparison of echocardiographic measurements from a control heart and an SRV heart at 33-weeks gestation.** Control (NML; a-1) and SRV (a-2) subject segmentations demonstrate identified boundary quality. Strain analysis (b-1) indicates comparable ventricular deformation in utero. Volume analysis (b-2) shows the SRV is in overload. Peak diastolic pressure fields (c-1) show a large vortex develops along the free-wall of the SRV, inducing greater energy loss (d-1). Late diastole pressure fields (c-2) and energy loss (d-2) behave similarly to early diastole. Timeseries curves are marked at early diastole and late diastole with gray lines. Inflow velocity (e-1) is comparable in utero. Intraventricular pressure difference ($\Delta P$) (e-2) shows elevated pressure recovery for the SRV heart. FEL measurements (e-3) show the SRV heart has a two-fold increase in loss across the field due to the free-wall vortex.

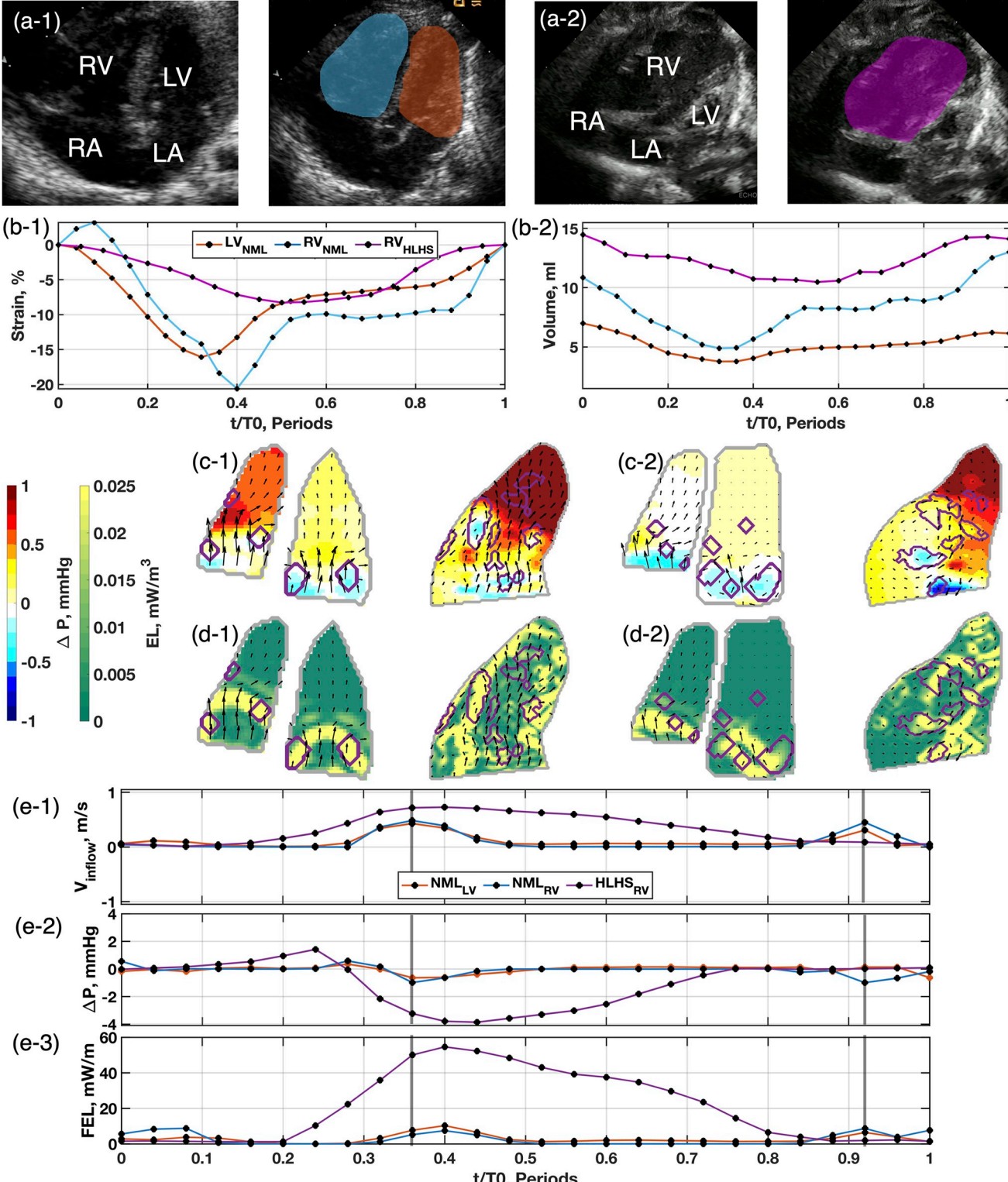

**Fig 3. Comparison of echocardiographic measurements from a control heart and an SRV heart at first week of birth.** Control (NML; a-1) and SRV (a-2) subject segmentations demonstrate identified boundary quality. Strain (b-1) is significantly reduced in the SRV. Volume analysis (b-2) shows the SRV is in overload. Peak diastolic pressure fields (c-1) show flow in the SRV is disordered, inducing greater energy loss (d-1). Late diastole pressure fields (c-2) and energy loss (d-2) behave similarly to early diastole. Timeseries curves are marked at early diastole and late diastole with gray lines. Inflow velocity (e-1) is comparable in magnitude, but the SRV inflow shows fused early and late diastole. Intraventricular pressure difference ($\Delta P$) (e-2) shows elevated pressure recovery for the SRV heart. FEL measurements (e-3) show the SRV heart has a nearly ten-fold increase in loss across the field.

observed during both diastole phases (Fig 3C-1 and 3C-2), where the free wall vortex had stronger low pressure and the apex had stronger high pressure compared to the healthy heart. Stronger EL was observed during both diastole phases (Fig 3D-1 and 3D-2) compared to the healthy heart due to more disordered flow which produces greater shear. The SRV timeseries (Fig 3C and 3D) did not showed fused diastole phases, stronger reversal IVPD and peak FEL compared to the healthy heart due to altered filling patterns.

## Discussion

Assessing diastolic function non-invasively in perinatal SRV patients remains challenging. Standard measurements from echo correlate poorly with diastolic function, especially in the perinatal transition [30], precluding the development of well-timed primary prevention strategies. This study explored the use of fetal and neonatal echo to quantify cardiac biomechanics, comparing standard and novel parameters of cardiac function to better understand perinatal changes in normal and SRV hearts. The work further demonstrates that novel hydrodynamic parameters, which can be quantified from scans collected during anatomy ultrasounds, can also detect functional changes in the SRV.

In SRV defects, the RV undergoes remodeling to support pulmonary and systemic circulations, reflected by quantified standard hemodynamic parameters using echo–SV and CO. SRV patients consistently exhibited larger SV (2.91 mL *in utero*, 3.70 mL *ex utero*) compared to the control RV (2.47 mL *in utero*, 2.59 mL *ex utero*) and the control LV (1.56 mL *in utero*, 2.84 mL *ex utero*). Similar trends were observed in CO, a function of SV and heart rate. This suggests that SRV enlargement is a response to meet blood volume requirements and begins during the fetal timeframe.

During the perinatal transition, standard diastolic function measurements showed little change. However, consistent significant differences were observed between SRV and control patients. SRV E/e' was consistently higher (17.5 *in utero*, 15.7 *ex utero*) than the control RV (8.8 *in utero*, 10.1 *ex utero*) and the control LV (8.5 *in utero*, 11.4 *ex utero*). This may also be associated with elevated IVPD required to empty and fill the SRV (Figs 1E-2 and 2E-2). Together, the SRV may require elevated filling pressures to overcome increased ventricular stiffness and reduced contractility.

The early and late diastole phases consistently appeared fused in SRV patient measurements (as illustrated in Figs 2E-1 and 3E-1). Prenatally, this fusion stems from the SRV needing to meet CO demands, which are affected by preload and afterload conditions. Postnatally, this altered filling pattern persists with the onset of spontaneous respirations and the predicted fall in pulmonary vascular resistance.

The increased SRV volume allows for altered diastolic flow, exhibited in larger vortex formations (Figs 2C and 3C), more disorganized flow (Figs 2D and 3D), and greater flow energy loss (Figs 2D and 3D). In healthy hearts, a donut-like vortex ring forms at the annular valve leaflet tips [31], which appears as two counter-rotating vortexes, as depicted in Fig 4. Normally, these vortices help aid in efficient ventricular filling [32]. In the SRV, the free wall vortex occupies a larger area because of the increased volume, resulting in greater flow energy loss, which reduces efficient flow redirection prior to systolic ejection. These observations are consistent with our recent study in Tetralogy of Fallot patients [15].

In single ventricle subtype SRV patients, such as hypoplastic left heart syndrome, the RV rapidly enters heart failure within the first week of life due to pulmonary over-circulation and systemic under-circulation. Treatments like Prostaglandin 1 (PGE 1) are administered to counter these effects, but patients are often quickly moved to palliative interventions (e.g., Norwood procedure or hybrid palliation). Patient progression to heart failure is inconsistent,

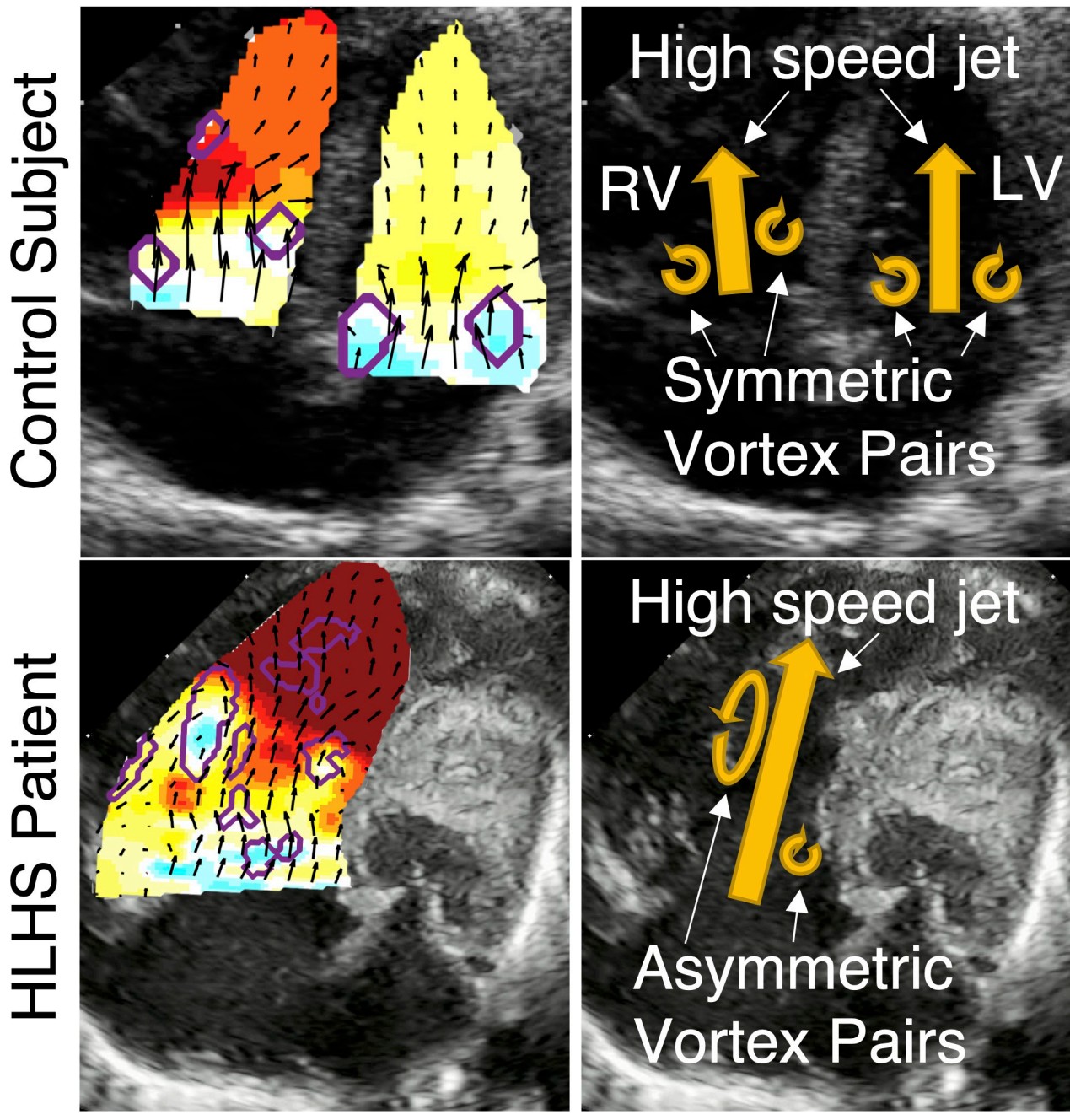

**Fig 4. Demonstration of the vortex pairs that form along the annular valve leaflet tips during diastolic filling.** In the healthy heart (top row), the pair that forms is nearly symmetric and acts as liquid rollers, helping push blood toward the apex to wash out each chamber. In the SRV patient (bottom), the pair is asymmetric, with a larger free wall vortex, which causes greater energy loss for proper filling and wash out of the chamber.

suggesting that the underlying mechanisms are not fully understood. The early period treatments and decisions are crucial due to rapid physiological changes, which are often not reliably captured by current medical imaging analysis. The methods described here could provide improved sensitivity to these physiological changes, offering a better understanding of ventricular work and optimizing patient management.

## Study limitations

Our study cohort size (10 SRV subjects) may not yet capture the statistics of the broader population. Furthermore, the study observed fetuses at 33-weeks gestational age, so the biomechanics quantified may not represent those immediately before birth. Moreover, the cohort comprises a diverse population of single ventricle subtype SRV subjects exhibiting varying hemodynamics and volume loading conditions. Given this diversity, drawing definitive conclusions regarding altered biomechanics and hemodynamics during the perinatal transition necessitates investigation in cohorts with stringent inclusion/exclusion criteria, focusing on individuals with more consistently similar findings. Additionally, the findings of this work cannot be applied to other SRV subtypes, such as transposition of the great arteries (TGA) or congenitally corrected TGA. These subtypes also have an SRV that sustains the systemic circulation before any surgical switch operation. These subtypes will also be considered in future studies when focusing on individuals with more consistently similar findings.

Imaging limitations in fetal echocardiography require novel measurement algorithms to be developed to ensure the most robust evaluation possible. Importantly, although the methods employed in this work have been demonstrated in prior studies, this is the first time the tools have been used to build a collective picture of fetal and neonatal biomechanics from echocardiography. However, because they have been developed for adult populations, they may not be directly applicable to fetal and neonatal echocardiograms without the need for further and extensive verification against gold standard measurements, which would include cardiac catheterization. One such instance is the algorithm used for annulus tracking to collect peak tissue motion measurements (s', e', a'), where frame rates can impact the ability to resolve measurements that match tissue Doppler imaging (TDI). Another instance is using the Simpson rule to compute volumes for the SRV hearts, which can be abnormally shaped. While volume estimates may hold clinical value, they may not represent the true volume, introducing greater uncertainty into subsequent calculations, such as ventricular mass [33].

We plan to pursue additional fetal and pediatric measurements in future studies with accompanying catheterization data to further verify the methods used in this work and to enable further quantification of functional differences. Furthermore, we plan to include additional measurements, such as global circumferential strain (GCS), which were not considered in this current study but have relevance when considering changes in SRV cardiac function. Finally, the implemented analysis method described here is automated but does require three user-selected initialization points. A fully automated method will be explored, where the three user-selected points will be replaced with three points found by AI-based feature detection tools.

## Conclusion

This work evaluated cardiac function biomarkers for SRV patients and age-matched controls from fetal and neonate echocardiograms. The methods used in this work collected conventional biomarkers routinely gathered during examination along with novel hemodynamic and hydrodynamic biomarkers derived from a new color Doppler reconstruction algorithm. Conventional biomarkers indicate that the SRV contracts and deforms like functionally normal biventricle hearts, even as SV and CO reflect the added volume taken on. Only through observing the novel biomarkers can functional changes during diastole in the SRV be observed. Importantly, these new biomarkers allow better quantification of myocardial performance, potentially improving the diagnosis and management of fetal heart failure. Altered hemodynamics and reduced ventricular relaxation were observed in the presence of a severe CHD, indicating the methods may provide earlier detection of anomalies *in utero* and lead to improving treatment practices *ex utero*.

## Supporting information

**S1 Text. Expanded description of the algorithms employed in the automated analysis platform.**
(DOCX)

**S1 Table. List of all abbreviations used in the manuscript.**
(DOCX)

**S2 Table. Complete list of echocardiographic measurements obtained from automated analysis platform.**
(DOCX)

**S1 File. Post-processing measurements from each subject used in the statistical analysis to compose Table 3 and S2 Table.**
(CSV)

## Author Contributions

**Conceptualization:** Brett A. Meyers, R. Mark Payne, Pavlos P. Vlachos.

**Data curation:** Brett A. Meyers, Yue-Hin Loke, R. Mark Payne.

**Formal analysis:** Brett A. Meyers, Sayantan Bhattacharya.

**Funding acquisition:** R. Mark Payne, Pavlos P. Vlachos.

**Investigation:** Brett A. Meyers, R. Mark Payne, Pavlos P. Vlachos.

**Methodology:** Brett A. Meyers, Yue-Hin Loke, R. Mark Payne, Pavlos P. Vlachos.

**Project administration:** Pavlos P. Vlachos.

**Resources:** Pavlos P. Vlachos.

**Software:** Brett A. Meyers, Pavlos P. Vlachos.

**Supervision:** Pavlos P. Vlachos.

**Visualization:** Brett A. Meyers, Sayantan Bhattacharya.

**Writing – original draft:** Brett A. Meyers.

**Writing – review & editing:** Brett A. Meyers, Sayantan Bhattacharya, Melissa C. Brindise, Yue-Hin Loke, R. Mark Payne, Pavlos P. Vlachos.

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
