## [Decision Letter · Decision Letter 0]

3 Jan 2024

PONE-D-23-30430Fetal and Neonatal Echocardiographic Analysis of Biomechanical Alterations for the Hypoplastic Left HeartPLOS ONE

Dear Dr. Vlachos,

Thank you for submitting your manuscript to PLOS ONE. After careful consideration, we feel that it has merit but does not fully meet PLOS ONE’s publication criteria as it currently stands. Therefore, we invite you to submit a revised version of the manuscript that addresses the points raised during the review process.

We look forward to receiving your revised manuscript.

Kind regards,

Hany Mahmoud Abo-Haded, MD

Academic Editor

PLOS ONE

Journal Requirements:

4. In this instance it seems there may be acceptable restrictions in place that prevent the public sharing of your minimal data. However, in line with our goal of ensuring long-term data availability to all interested researchers, PLOS’ Data Policy states that authors cannot be the sole named individuals responsible for ensuring data access (http://journals.plos.org/plosone/s/data-availability#loc-acceptable-data-sharing-methods).

Additional Editor Comments:

This is a small study exploring the biomechanics of the single right ventricle during perinatal transition, particularly focussing on novel measures.

In my opinion, this study needs to be reframed as a technical paper. The study is too small and the variety of patient anatomy too great to “understand the fetal and neonatal SRV..” as stated in the abstract.

I think if reframed, this could be an interesting and useful technical paper.

Reviewers' comments:

Reviewer's Responses to Questions

**Comments to the Author**

1. Is the manuscript technically sound, and do the data support the conclusions?

Reviewer #1: Yes

Reviewer #2: Yes

Reviewer #3: Partly

Reviewer #4: Partly

2. Has the statistical analysis been performed appropriately and rigorously? 

Reviewer #1: Yes

Reviewer #2: Yes

Reviewer #3: Yes

Reviewer #4: Yes

3. Have the authors made all data underlying the findings in their manuscript fully available?

Reviewer #1: Yes

Reviewer #2: Yes

Reviewer #3: No

Reviewer #4: Yes

4. Is the manuscript presented in an intelligible fashion and written in standard English?

Reviewer #1: Yes

Reviewer #2: Yes

Reviewer #3: Yes

Reviewer #4: Yes

5. Review Comments to the Author

Reviewer #1: I congratulate with authors for their novel type of echocardiography examination.

The manuscript is well written in each section and well structured with appropriate attention for style!

As an Obstetrician involved in prenatal ultrasound diagnosis I dont have the appropriate background to evaluate the mathematics calculation behind the formulas. Images and Table are of high quality.

Reviewer #2: Thank you for providing the opportunity to review the paper titled 'Fetal and Neonatal Echocardiographic Analysis of Biomechanical Alterations for the Systemic Right Ventricle Heart.' In this paper, the authors reported on changes in diastolic function of the SRV using a novel ultrasonographic method during both the prenatal and postnatal periods. The manuscript was well-written, and I have no concerns regarding this paper.

Reviewer #3: Title: Fetal and Neonatal Echocardiographic Analysis of Biomechanical Alterations for the Hypoplastic Left Heart

PONE-D-23-30430

This is a small study exploring the biomechanics of the single right ventricle during perinatal transition, particularly focussing on novel measures.

In my opinion, this study needs to be reframed as a technical paper. The study is too small and the variety of patient anatomy too great to “understand the fetal and neonatal SRV..” as stated in the abstract.

I think if reframed, this could be an interesting and useful technical paper.

Abstract

The background could better reflect the study. The objective of the study seems to be to compare standard echo measures to novel measures using a range of SRV case types. In my opinion, the study is largely a technical paper about use of novel biomechanical measures in a small group of patients.

Introduction

The introduction lacks cohesion and after reading it, I was still not clear what this study was about. It does not address a) perinatal transition or b) neonatal hemodynamics of the SRV. I would encourage the authors to clarify what the study is really about: that hemodynamic changes in the SRV during perinatal transition are poorly understood, and that current measures may insufficiently capture systolic and diastolic function. Then, the aims of the study are to compare novel and standard measures to better describe perinatal changes.

Line 66 – 70 discuss benefits of fetal detection and that pediatric echo is used to treat. Whilst true, these statements aren’t really relevant to the paper.

Line 71 – remove “i.e.”. SV and CO are not the only measures of systolic function and are not routinely reported in most centres. It would be better to say “e.g.” and quote measures more commonly used, such as RVFAC, TAPSE, TDI S’.

Line 77 – what do the authors mean by “hemodynamics” here? This is too vague; I am genuinely uncertain what they are referring to, and the reference is to an overall guideline from 2004. As an aside, there are several more recent AHA / JASE guidelines on Fetal Echo – the Donofrio 2014 one is quoted. Moon-Grady 2023 should also be quoted: https://doi.org/10.1016/j.echo.2023.04.014.

Line 79 – 82: I think this paragraph needs rewording. Software does not rely on automated segmentation to work, as the workflow is to manually trace the chambers. Perhaps “rely” is not the correct word? The discussion regarding segmentation is unclear, and not well referenced – only to overall guidelines. Do they mean segmentation of the image into an LV / RV for example? I think this is what they mean, and if better defined, it will help to clarify the other points. This is indeed manual. Or segmentation of the ventricles? This is done automatically by software (Tomtec, EchoPac, Syngo). Strain is not difficult to obtain, for example, a global longitudinal strain measurement on a fetal study. Strain does not “rely” on ventricular segmentation to my knowledge, although of course it relies on segmentation of the image (identification of the ventricle). This is a somewhat superficial point in a paper which is not really about automatically finding parts of the image (e.g. through an AI algorithm). The authors approach also relies upon manual identification of the annulus and apex. It is true that dividing the ventricle, particularly a SRV, into, for example, a six-segment model, is difficult, and whether the septum should be included or not (as it constitutes a variable amount of wall) in a HLHS fetus / infant is unclear.

Methods

The methodology is largely a detailed mathematical description of analysis. This is beyond my ability to review critically.

It is unclear what frame rates the images were taken at or stored at. Given the rest of the analysis is likely frame-rate dependent, and the high heart rates in fetuses and neonates, this should be quoted.

Line 112: I believe the machine was an SC2000 – there is a zero missing.

The authors state that automated speckle tracking of the mitral annulus has been previously validated. Indeed, speckle tracking includes the annulus in general. However, as the authors spell out at length the process for tracking the annulus, is the algorithm new in some way? If so, how has it been validated? If not different from standard speckle tracking, what is new?

The peak annulus velocities in standard TDI differ from this process, as no tracking is performed – it is derived from movement of the annulus THROUGH a static pulse-wave sector. Therefore, the normal values are likely different. Is there prior data defining a normal range in fetuses and newborns, and how it compares to standard TDI? The reference [21] refer to accuracy using adult validation – much lower heart rates. You cannot extrapolate.

The reference [25, 2020] for vortices seems to be theoretical and small animal models. It there a follow-up in humans? What would be considered the gold standard? It seems that the authors have moved strait to assessing pathology using this method.

Line 149: what limitations are the authors referring to – this is not clear.

Line 151: GLSr is not defined at first use, and not a standard abbreviation.

Line 184: There is a sentence fragment.

Line 188: Segmentation using Simpson’s rule – do the authors mean calculation of volume based on length and cross-sectional ellipses? Perhaps this needs to referenced (if one exists) as accurate for a single right ventricle, which is not conical?

No explanation on how the SV / CO, or many of the other standard echo measures were obtained is reported. These can be done using geometric assumptions or the continuity equation. This should be specified.

Results

Restrictive patent foramen ovale is not a subtype of SRV – this needs clarification. Surely, as the authors have the images, the subtype should be able to be reported on all the studies – one is not.

Two of the patients have L-looped ventricles, one cardiogenetic shock as a complication, one total anomalous veins and one moderate TR! This is a highly heterogenous group hemodynamically. I don’t think you can put them all together and draw any conclusions about the SRV. This loops back to the paper being largely technical – the differences in data that are obtained from the novel methodologies may be useful, even in a heterogenous group. If the authors disagree, further anatomical description / explanation to justify this is required.

In reporting, more clarity on which results (fetal vs newborn) are being reported is needed.

The authors reference and interpret diastolic parameters using guidelines which, to my knowledge, are not validated, and have evidence that they do not necessarily reflect dysfunction in either fetuses, neonates or SRV. I recommend that they refrain from interpretation, and just describe the parameters they evaluated. Ref 20 is not the most recent and is focussed on adults.

Do the authors have any data on correlation between measures such as IVPD and E/e’?

Discussion

Lines 292 – 319 is largely data that is already known about the SRV using standard echo techniques and could be more efficiently discussed.

The discussion of diastolic function is logically flawed – the authors first note that “conventional imaging measures do not correlate with diastolic function”, but then use the same measures to make conclusions, for example, that the filling pressures are elevated. The E/e’ in a fetus / neonate with altered filling conditions and afterload may not mean the same thing as in an adult or even a normal neonate. Again, the reference is to a document focussed on adults. And not for the right ventricle. The evidence quoted from Line 328 is fairly weak – the MPI is not a strong measure of DD. More reliance on atrial contraction can be due to higher volume requirements. Without a measure of wall stress / filling pressures, the impact on the ventricular muscle in terms of pathways leading to fibrosis are unknown.

Line 315 319: The phases of inflow are explained as due to exposure to systemic and pulmonary pressure. This is likely not the reason and requires clarification. The SRV in utero is exposed to higher diastolic volume loading and required to produce higher cardiac output. The preload will depend on the compliance of the ventricle (it may be able to maintain low EDP), and the afterload on systolic function, placental function and SVR. The pulmonary pressures should not influence these parameters, and should not influence the relative phases of filling. Filling further increases after birth with falling PVR, so 318 – 319 is likely true.

Line 340 - Should this be in the limitations section? While it’s unclear what the “above methods” refers to, if the authors refer to the methods they used in this study, I agree that they are not directly applicable without better validation. These techniques need direct correlations with gold standard measures such as invasive animal models or cath lab Ees measures of function before we can draw conclusions. If this data in fact exists, this needs to be elucidated in the discussion and methods.

Limitations: The study is retrospective, with small numbers. The fetal studies were quite early (33 weeks), which means they may not be representative of biomechanics immediately prior to birth.

Reviewer #4: The manuscript entitled "Fetal and Neonatal Echocardiographic Analysis of Biomechanical Alterations for the

Hypoplastic Left Heart" uses a novel imaging algorithm to quantify measurements of fetal and neonatal echocardiography in healthy and SRV hearts. The data show not only early diastolic dysfunction in SRV hears compared to healthy controls, but also some subtle differences in systolic function during the neonatal period. While the image analysis is intricately detailed in the methods, this imaging algorithm to my knowledge is an n=1. One question that I have, is the imaging algorithm used in Matlab for this analysis available on github or some other public forum for other users. This is important as the measurements detailed herein would only be acquired by the sites involved in this particular study and would not be generalizable to a broader user, in particular neonatologists and pediatric cardiologists in a clinical setting.

One of the major concerns in this manuscript is that the data acquired from this algorithm is not compared to any other clinical measurement for the purpose of validation and there is no way to assess whether these measurements are of clinical quality or not. Given that this was a retrospective study, it would seem reasonable that these echoes would have been quantified by health care practitioners and would likely have this data stored in the electronic medical records as a way to compare the algorithm measures of standard measurements such as E, A, E/e', etc.. as a way to compare those standard measurements against the algorithm. Without some way of comparing the algorithm to a gold standard we are only left to assume these measurements are valid. If possible, can the authors provide some comparisons between standard software derived measures against the algorithm.

Also, from the statistical analysis the authors used a students t-test to compare differences between groups in this analysis. Since you used a parametric analysis, why weren't the reported values as mean (SD)? Was your data normally distributed and did the authors perform a normality test prior to the use of a parametric test. The authors reported median (IQR) which is fine but usually these values are reported when using non-parametric tests like the Wilcoxon.

Below I have some minor comments:

1) Line 185: There is a sentence that consists of "This." I presume that is a typo, please remove.

2) Line 205: the prhase "to under differences in filling", I presume you meant understand differences in filling

3) Line 236: The authors say that SV is increased compared to control LV and RV in the prenatal setting but the statistics would suggest that SV in SRV is increased compared to the LV and trending in the RV (p=0.06)

4) Line 275: I believe the reference was to the E/A ratio but in the sentence, it says E/, please fix this for clarity

5) Line 287: "Stronger EL was observed both diastole", might want to add the during after observed

6) Line 311: The median E/e' in SRV group during postnatal was listed as 1.57, which is a typo based on the table showing 15.7.

6. PLOS authors have the option to publish the peer review history of their article (what does this mean?). If published, this will include your full peer review and any attached files.

Reviewer #1: No

Reviewer #2: **Yes: **Katsusuke Ozawa

Reviewer #3: No

Reviewer #4: No

---

## [Author Response · Author response to Decision Letter 0]

16 Feb 2024

Per the editor's request (addressed in the updated cover letter):

1) The revised manuscript agrees with PLOS One formatting requirements.

2) We have reviewed the note from Dr. Chenette and Mr. Hrynaszkiewicz, and understand that posting the raw data with a repository may increase citation rate. We cannot provide the raw image files or resulting measurement fields associated. However, we can and will provide our measurements used to generate the values reported in Table 3 (previously Table 2 in original submission).

3) We have reviewed the journal guidelines pertaining to code sharing. The algorithms used in this study are patented and/or copyright protected. We therefore cannot share these in an open-source format. We have added the following statement to the manuscript for Data Availability,

BAM and PPV have utility patents and/or copyright protections for each analysis algorithm used in this manuscript. Codes can be shared after proper licensing is obtained through the Purdue Office of Technology Commercialization. 

4) Upon further review, a minimum dataset of quantities used to generate the plots and supplemental tables in this study can be provided. This minimal anonymized dataset has been provided as supplemental material.

All additional responses to reviewer and editor comments is provided in the "Response to Reviewers" letter.

---

## [Decision Letter · Decision Letter 1]

3 Jun 2024

PONE-D-23-30430R1Fetal and Neonatal Echocardiographic Analysis of Biomechanical Alterations for the Hypoplastic Left HeartPLOS ONE

Dear Dr. Vlachos,

Thank you for submitting your manuscript to PLOS ONE. After careful consideration, we feel that it has merit but does not fully meet PLOS ONE’s publication criteria as it currently stands. Therefore, we invite you to submit a revised version of the manuscript that addresses the points raised during the review process.

Academic Editor response to authors:

The authors are possibly trying to answer the wrong question or using the wrong comparisons to answer it.

We dont really know what predestines some systemic RVs to fail while  others do well, and I suspect that is what is behind a study like this where standard echo oarameters are not sensitive enough... But I think this study may have within it the data which may start to unravel this...

The fetal systemic RV is doing much the same job as it is postnatally (in a 'single ventricle' circulation, whereas there are much larger changes in the loading on the systemic LV (or indeed the subpumonary RV) in the biventricular circualation. THus using the normal hearts as controls is counterintuitive at least to me. To me the 'money' is in the changes from pre to postnatal parameters on the same ventricle, +/- the same ventricle in a biventriclar ciurculation eg ccTGA, or indeed TGA pre repair. Comparing the CHANGES to the changes for the biventricular LV may also be of interest but there are much greater differences in the loading conditions in this especially depending on timing of ductal closure as well as the timing of the drop in PVR.. which varies per individual.

So I think possibly reanalysing these data in a different way would be of more physiologic interest. I realy dont thing comparing absolue measurements of any of these parameters between different ventrciular morpholgies makes much sense as intrinsically they are not only constructed differently at the cellular and fibromuscular architecture level but they have different shapes, different loading conditions and even different interventricular interactions depending on the status of the other ventricle

So to me this study demonstrates some novel techniques but the questions as to their utility remain, to me as yet, completely unaddressed. Maybe I have missed the point but I dont think this is publishable without some reference to this.

We look forward to receiving your revised manuscript.

Kind regards,

Hany Mahmoud Abo-Haded, MD

Academic Editor

PLOS ONE

Reviewers' comments:

Reviewer's Responses to Questions

**Comments to the Author**

1. If the authors have adequately addressed your comments raised in a previous round of review and you feel that this manuscript is now acceptable for publication, you may indicate that here to bypass the “Comments to the Author” section, enter your conflict of interest statement in the “Confidential to Editor” section, and submit your "Accept" recommendation.

Reviewer #5: (No Response)

Reviewer #6: (No Response)

2. Is the manuscript technically sound, and do the data support the conclusions?

Reviewer #5: Partly

Reviewer #6: Yes

3. Has the statistical analysis been performed appropriately and rigorously? 

Reviewer #5: I Don't Know

Reviewer #6: Yes

4. Have the authors made all data underlying the findings in their manuscript fully available?

Reviewer #5: Yes

Reviewer #6: Yes

5. Is the manuscript presented in an intelligible fashion and written in standard English?

Reviewer #5: Yes

Reviewer #6: Yes

6. Review Comments to the Author

Reviewer #5: The authors compared echocardiograhpy-derived biomehcanic data of 10 systemic right ventricels subjects with that of normla control subjects in both prenatal and postnatal period. and they showed increased stroke volume and cardiac output but also increase neergy loss, reduced diastolic mechanics in those with systmeic RV.

First, the authors work is novel and do demonstrated increased understanding of altered cardiac mechanics by using advance echocardiographic technique. This may help to increase our understanding of the pathophysiology.

However, the comparison between the patients and controls is compounded by the fact that the patients had unbalanced ventricles. The right ventricle in the subjects is not only pressured loaded when compared to normal subpulmonary ventricle, but also volumed loaded because the systemic RV has to provide driving force to both pulmonary and systemic circulation, which is unlike in other systemic RV (such as TGA or CCTGA) that the systemic RV only sustains systemic circulation.

As such their findings have to be interpreted with caution and could not be applied to TGA or CCTGA, both also having systemic RV before surgical switch operation. And this should be described and addressed in their discussion that they are studying a subset of systemic RV.

Reviewer #6: This paper is well-structured, with detailed results, and a complete and comprehensive discussion, also acknowledging the study's limitations. The data presented are original and interesting, and the authors have clearly emphasised the lack of data in this field and the potential benefit and knowledge that their paper could provide.

Some paragraphs are challenging for some reader that has no strong familiarity with the algorithm beyond some of the measurements adopted. A more easy explanation of the novel hydrodynamic parameters would enhance the reader's understanding. Especially the image analysis workflow section [130-248] maybe should be simplified and further summarized with the help of more diagrams or figures.

The results are well presented with appropriate use of tables and figures to illustrate key findings. Maybe the clinical relevance of the findings should be further stressed. The potential clinical applications of the novel parameters are generic and should be better articulated in the conclusion. Highlighting the clinical implications of these findings more explicitly would strengthen the impact of the results.

In summary, the paper is well-written and provides valuable datas. With minor revisions, it would be a fit for publication.

7. PLOS authors have the option to publish the peer review history of their article (what does this mean?). If published, this will include your full peer review and any attached files.

Reviewer #5: No

Reviewer #6: No

---

## [Author Response · Author response to Decision Letter 1]

17 Jul 2024

Response to reviewer and editor comments have been provided in an attached Response To Reviewers document.

---

## [Editor Report · Decision Letter 2]

29 Jul 2024

Fetal and Neonatal Echocardiographic Analysis of Biomechanical Alterations for the Systemic Right Ventricle Heart

PONE-D-23-30430R2

Dear Dr. Vlachos,

We’re pleased to inform you that your manuscript has been judged scientifically suitable for publication and will be formally accepted for publication once it meets all outstanding technical requirements.

Kind regards,

Hany Mahmoud Abo-Haded, MD

Academic Editor

PLOS ONE
---

## [Editor Report · Acceptance letter]

2 Aug 2024

PONE-D-23-30430R2 

PLOS ONE

Dear Dr. Vlachos, 

I'm pleased to inform you that your manuscript has been deemed suitable for publication in PLOS ONE. Congratulations! Your manuscript is now being handed over to our production team.

Kind regards, 

on behalf of

Professor Hany Mahmoud Abo-Haded 

Academic Editor

PLOS ONE